# Thermogenic Differentiation of Human Adipocyte Precursors in Culture: A Systematic Review

**DOI:** 10.3390/cells14231907

**Published:** 2025-12-02

**Authors:** Gislainy Lorrany Anatildes da Silva de Paula, Erica Correia Garcia, Bruna Teles Soares Beserra, Angelica Amorim Amato

**Affiliations:** 1Laboratory of Molecular Pharmacology, Department of Pharmaceutical, Sciences University of Brasilia, Brasilia 70910-900, Brazil; gislorrany@gmail.com (G.L.A.d.S.d.P.); erica.correia.garcia@gmail.com (E.C.G.); 2Food and Nutrition Department, School of Nutrition, Federal University of Mato Grosso, Cuiabá 78060-900, Brazil; brunna.tsb@gmail.com

**Keywords:** adipocyte differentiation, cell culture, thermogenic adipocyte

## Abstract

Thermogenic adipocytes present a promising therapeutic strategy for metabolic diseases. While murine models have provided valuable insights into thermogenic adipose tissue, their relevance to human physiology is constrained by species-specific differences in tissue distribution and thermogenic capacity. In vitro human models offer a more controlled platform to study adipocyte differentiation, addressing challenges such as limited access to deep fat depots and individual variability. This systematic review summarizes the current literature on human in vitro models for thermogenic adipocyte induction, encompassing 117 studies involving primary human adipocyte progenitors differentiated into thermogenic adipocytes in 2D cultures. Most studies relied on classical adipogenic inducers, including isomethylbutylxanthine, dexamethasone, and insulin, with additional use of triiodothyronine, rosiglitazone, or indomethacin. A few studies incorporated adrenergic stimulation or exposure to lower temperatures to simulate cold exposure. Notably, some studies demonstrated successful differentiation under serum-free, chemically defined conditions, highlighting their potential for reproducibility and translational relevance. A key limitation remains the predominant reliance on gene expression as the primary outcome, with few studies assessing mitochondrial respiration or broader metabolic functions. Moving forward, the development and adoption of standardized, functionally validated protocols will be critical to fully realize the potential of human in vitro thermogenic adipocyte models in metabolic research.

## 1. Introduction

Obesity is a chronic and multifactorial disease defined by the excessive accumulation of body fat; it significantly increases the risk of type 2 diabetes, cardiovascular disease, certain cancers, and overall mortality [1]. Currently available treatment strategies for obesity primarily aim to reduce energy intake [2], with relatively limited emphasis on enhancing energy expenditure. Lifestyle interventions—particularly hypocaloric diets, macronutrient modulation, and intermittent fasting—remain the foundation of clinical management, producing initial weight loss through sustained caloric restriction [2]. Pharmacotherapies such as glucagon-like peptide 1 receptor agonists reinforce the intake-focused paradigm by reducing appetite and slowing gastric emptying, with no proven direct effects on energy expenditure [3]. Bariatric surgery, the most effective long-term treatment for severe obesity, also acts chiefly by restricting intake, although some procedures may modestly influence energy expenditure and substrate utilization [2].

The prevailing focus on intake restriction underscores a critical gap in current therapies and highlights the need for strategies that effectively and safely enhance energy output in parallel with intake modulation. While physical activity contributes to increased energy output, its impact on weight loss is typically modest [4]. An alternative or synergistic approach involves stimulating nonshivering thermogenesis in thermogenic adipose tissue, which comprises brown and beige adipocytes [5]. Brown adipocytes are typically located in dedicated depots, such as the deep neck and supraclavicular regions, and are present from infancy through adulthood [5]. In contrast, beige adipocytes emerge within white adipose tissue in response to specific stimuli, including cold exposure and β-adrenergic activation [5]. Both cell types contribute to increased energy expenditure and have emerged as potential therapeutic targets for obesity and metabolic disorders. However, the extent, inducibility, and metabolic relevance of beige adipocytes in adult humans remain areas of active investigation.

In this scenario and considering the substantial challenges associated with investigating human thermogenic adipocytes in vivo—such as limited accessibility of deep fat depots, variability in thermogenic capacity across individuals, and the influence of environmental and physiological factors [6]—human cellular models have emerged as valuable platforms for both mechanistic studies and translational research, enabling controlled investigation of the molecular pathways underlying beige adipocyte differentiation and function. Therefore, we conducted a systematic review to critically evaluate and synthesize the existing literature on human in vitro models for thermogenic adipocyte induction.

## 2. Materials and Methods

### 2.1. Statement and Registration

This systematic review followed the recommendations of the Preferred Reporting Items for Systematic reviews and Meta-Analysis (PRISMA) [7] and was registered on PROSPERO (CRD420251019278). The PRISMA checklist is presented in Appendix A.

### 2.2. Search Strategy

The literature search strategy was developed using the PICOS acronym, in which P (population) was defined as ‘human adipocyte precursors,’ I (intervention) as ‘exposure to differentiation medium to induce the differentiation of thermogenic adipocytes,’ C (comparator) as ‘comparison group with a control group (non-differentiated adipocyte precursors, adipocyte precursors induced to differentiate into white adipocyte, or precursors from different adipose depots),’ O (outcome) as ‘molecular markers of thermogenic adipocytes (mRNA, protein), mitochondrial content, or functional measures of oxygen consumption,’ and S (study type) as ‘cell culture-based studies assessing the differentiation of adipocyte precursors into thermogenic adipocytes.’

We searched PubMed, Web of Science, Scopus, and Google Scholar from inception to 26 March 2025, without language restriction, using the search strategy detailed in Appendix A. The reference lists of included articles were also manually searched. References were managed using a reference management system [8].

### 2.3. Eligibility Criteria, Study Selection, and Data Extraction

The studies were selected in a two-phase process. In the first phase, two reviewers (G.L.A.S.P. and E.C.G.) independently screened the retrieved studies by assessing their titles and abstracts to select eligible studies. On the second phase, the same reviewers independently assessed the full-text version of eligible studies and applied the inclusion criteria. Disagreements between the reviewers in the first two phases were resolved through discussion with a third reviewer (A.A.A.).

Studies describing protocols to induce the differentiation of precursors into thermogenic adipocytes in primary cell models and assessing molecular markers or functional measures of the thermogenic adipocyte were included. While immortalized cell lines are widely used for mechanistic studies due to their ease of use, reproducibility, and scalability, their adipogenic potential—and likely their thermogenic capacity—may differ substantially from that of primary human cells [9]. To minimize confounding factors and enhance translational relevance, we therefore restricted our review to studies employing primary human adipocyte precursors.

Abstracts, reviews, editorials, book chapters, and studies that discussed mouse cells or cell lines or in which the composition of the differentiation medium was not described were excluded. When the composition of the medium used to induce the thermogenic phenotype was described in a citation, the citation was searched to retrieve the information. The following information were collected from the included studies: author, year of the publication, type of adipocyte precursor used, growth medium, cell confluency at adipocyte differentiation induction, composition of the medium used to induce differentiation into thermogenic adipocytes, and outcomes related to the thermogenic phenotype, including mRNA and protein expression of thermogenic markers, mitochondrial content, and oxygen consumption rate.

### 2.4. Quality Appraisal

The risk of bias was assessed independently by two reviewers (G.L.A.S.P. and E.C.G.) using the United States National Toxicology Program’s Office of Health Assessment and Translation (NTP/OHAT) risk of bias rating tool [10], and disagreements were resolved by discussion with a third reviewer (A.A.A.). Risk of bias was classified as definitely low (++), probably low (+), probably high (-), definitely high (--), and not applicable (NA).

### 2.5. Qualitative Analysis

We summarized data by describing the adipocyte differentiation protocols and results concerning characterization of the thermogenic adipocyte in a table format (structured summary method).

## 3. Results

### 3.1. Study Selection

A total of 6426 studies were identified in the searched databases. After duplicate removal, 2634 articles were screened and 270 were selected for full-text analysis. A total of 117 studies [11,12,13,14,15,16,17,18,19,20,21,22,23,24,25,26,27,28,29,30,31,32,33,34,35,36,37,38,39,40,41,42,43,44,45,46,47,48,49,50,51,52,53,54,55,56,57,58,59,60,61,62,63,64,65,66,67,68,69,70,71,72,73,74,75,76,77,78,79,80,81,82,83,84,85,86,87,88,89,90,91,92,93,94,95,96,97,98,99,100,101,102,103,104,105,106,107,108,109,110,111,112,113,114,115,116,117,118,119,120,121,122,123,124,125,126,127] were included (Figure 1), and their characteristics are described in Appendix A. The excluded studies and reasons for their exclusion are presented in Appendix A.

### 3.2. Characteristics of the Protocols to Induce Differentiation of Adipocyte Precursors into Thermogenic Adipocytes

The protocols described in the included studies to induce differentiation of primary adipocyte precursors into thermogenic adipocytes varied with respect to the specific precursor used, cell confluency at adipogenic induction, induction medium, and duration of induction (Appendix A). Most studies (n = 70/116) used cells from the stromal vascular fraction of adipose tissue obtained from different tissue depots, including subcutaneous white adipose tissue (abdominal, thigh, neck, supraclavicular, breast, subacromial, chin, knee, flank, hip), visceral white adipose tissue (omental, peri-carotid, pericardial, mediastinal, perirenal, periadrenal), deep neck adipose tissue, prevertebral adipose tissue, or fetal brown adipose tissue. A total of 35 studies reported that FGF was used during precursor expansion, before adipocyte differentiation induction (Table 1).

In most studies (n = 48/117), precursor cells were induced to differentiate into thermogenic adipocytes at 100% confluency. Fewer studies induced cells one (n = 1/117) to two days (n = 26/117) after confluency or at 70 to 95% confluency (n = 15/117). A total of 28 studies did not report confluency at adipogenic induction. Most studies (n = 112/117) reported the cell culture medium used to induce adipogenic differentiation, and the most frequently used were DMEM/F12 (n = 42/112), DMEM (n = 41/112), and DMEM/F12/Ham’s (n = 20/112). Most studies (n = 74/112) supplemented induction medium with serum, whereas in 38/112 studies, adipogenic differentiation was carried out in serum-free medium. The most frequently used serum was fetal bovine serum (38/74), with concentrations ranging from 2 to 20% (volume/volume). Adipocyte differentiation duration ranged from three to 32 days, with most studies conducting differentiation for 7–14 days. Moreover, in most studies (65/117) differentiation was conducted in two stages, comprising the addition of different substances to cell culture medium in each (Table 1).

All studies induced adipocyte differentiation using IBMX and dexamethasone, at concentrations ranging from 0.03 to 1 mM and 0.1 to 100 μM, respectively. Most studies (114/117) used insulin at concentrations ranging from 1 to 1000 nM, triiodothyronine (T3, 90/117) at concentrations ranging from 0.2 to 1000 nM, and rosiglitazone (89/117) at concentrations ranging from 0.1 to 10 μM. Few studies used indomethacin (30/117) at concentrations ranging from 2 to 100 μM. Other browning agents, such as bone morphogenetic proteins, fibroblast growth factors, irisin, and its precursor FNDC5 were used in a minority of the studies. In addition, transferrin, biotin, and pantothenate were added to the differentiation induction medium by less than half of the studies. Additionally, thermogenic induction was implemented by acute (<12 h) or chronic (>12 h) adrenergic stimulation in a subset of studies, whereas a smaller proportion employed exposure to lower temperatures for variable durations to upregulate thermogenic markers (Table 2).

### 3.3. Characteristics of Thermogenic Adipocytes in the Included Studies

The thermogenic phenotype was characterized by RNA (mRNA or miRNA, 104/117) and protein (70/117) of thermogenic markers, mitochondrial bioenergetics (48/117), or mitochondrial content (20/117). The thermogenic markers (RNA and protein) assessed in the included studies are presented in Figure 2, Table 3 and Appendix A.

Most studies (108 of 117) compared adipocyte precursors induced to differentiate thermogenic adipocytes with various control conditions, including undifferentiated precursors (25 of 108), precursors differentiated into white adipocytes (43 of 108), mature adipocytes not stimulated with adrenergic agents, forskolin or cAMP (28 of 108), precursors from distinct adipose depots (19 of 108), those differentiated in the absence of glitazones (9 of 108), or cells differentiated at lower versus higher temperatures (6 of 108). Among studies comparing the thermogenic potential of precursors derived from different anatomical depots, heterogeneous methodologies precluded a definitive synthesis. Nonetheless, a consistent trend was observed: twelve out of twelve studies reported greater thermogenic capacity in precursors from adult adipose tissue located in the deep neck, supraclavicular, perirenal, or periadrenal regions compared with those from superficial subcutaneous or visceral depots. Additionally, three out of four studies comparing developmental origins found that precursors from fetal tissues exhibited higher thermogenic potential than those derived from adult adipose tissue.

In most cases, the intended thermogenic phenotype was confirmed, based on mRNA and protein expression of thermogenic markers, mitochondrial content, and mitochondrial activity. Only a minority of studies assessed mitochondrial activity and content directly. Notably, among the 38 studies investigating thermogenic adipocyte differentiation under serum-free conditions, the majority reported successful differentiation, as evidenced by the induction of thermogenic mRNA expression (35 of 36 studies), protein-level confirmation (25 of 26), increased mitochondrial content (8 of 9), and enhanced mitochondrial bioenergetic function (17 of 18).

### 3.4. Risk of Bias in Individual Studies

The risk of bias was assessed using the US NTP/OHAT Risk of Bias Rating Tool for Human and Animal Studies, and the results are summarized in Appendix A. All included studies were classified as having a low risk of bias.

## 4. Discussion

Thermogenic adipose tissue, encompassing classical brown and inducible beige fat, has emerged as a compelling target for the treatment of obesity and associated metabolic disease [128]. Recent studies in rodents have highlighted the potential of activating or expanding thermogenic adipocytes to counteract excess adiposity and improve metabolic outcomes such as insulin sensitivity and glucose tolerance [128]. In this scenario, therapeutic strategies aimed at harnessing the metabolic capacity of thermogenic fat through pharmacological, environmental, or nutritional means are gaining traction as viable interventions in the fight against metabolic disease.

Despite notable physiological and molecular parallels between rodent and human adipose tissues, key species-specific differences in depot distribution, gene expression, and thermogenic regulation limit the direct translational applicability of rodent findings to human biology. In rodents, classical BAT is abundant and organized in discrete depots, while beige adipocytes are readily recruited within multiple WAT depots. In contrast, genuine BAT is primarily present during fetal and neonatal stages in humans, and thermogenic fat in adults is more restricted, residing mainly in the supraclavicular and thoracic regions. These depots exhibit molecular signatures more akin to murine beige adipocytes than classical BAT [129].

Although human and mouse thermogenic adipocytes share comparable UCP1-dependent mitochondrial function, the upstream regulatory mechanisms differ significantly. In mice, β_3_-adrenergic receptors are abundantly expressed and play a central role in activating thermogenesis and lipolysis. In humans, however, thermogenic adipocytes respond to both β_2_- [130] and β_3_ [39]-adrenergic stimulation, but with less pronounced effects, reflecting differences in receptor expression patterns and downstream signaling. Additionally, retinoic acid receptor signaling robustly induces UCP1 expression in murine adipocytes but fails to do so in their human counterparts [120], underscoring further divergence in transcriptional regulation of thermogenesis.

Although the study of human thermogenic adipocytes is critical for advancing translational strategies, in vivo investigation remains challenging due to the relative scarcity of thermogenic adipose tissue and its dispersed anatomical localization compared to white adipose depots. Therefore, the development of human cell models remains crucial for advancing our understanding of thermogenic adipose biology and its therapeutic potential. Given the importance of these models, we conducted a comprehensive literature review of studies investigating the differentiation of primary adipocyte precursors into thermogenic adipocytes in 2D culture systems, irrespective of the specific research question, to evaluate the differentiation protocols and the criteria used to define the thermogenic phenotype.

We identified 117 studies fulfilling our inclusion criteria, the majority (114/117) published after 2009, following the discovery that adult humans possess thermogenic adipose tissue and that its presence is positively correlated with metabolic health [131,132,133]. This surge in research aligns with growing recognition that human cell models, particularly those derived from primary tissues or pluripotent stem cells, offer a more physiologically relevant platform for elucidating the regulatory mechanisms of human thermogenic adipocytes and for screening pharmacological agents with translational potential.

Almost all studies induced adipocyte differentiation by exposing precursors to the classic adipogenic cocktail comprising the phosphodiesterase inhibitor isomethylbutylxanthine (IBMX), the synthetic glucocorticoid dexamethasone, and insulin [134], albeit with varying concentrations and exposure duration. IBMX leads to increased intracellular levels of cAMP and activation of protein kinase and C/EBP-δ. By activating glucocorticoid receptor, dexamethasone induces the expression of C/EBP-β. The latter, in concert with C/EBP-δ, induces the expression of C/EBPα and PPARγ. Insulin, in turn, induces transcription factors such as SREBP-1C to stimulate adipogenesis and lipogenesis [135].

A significant proportion of studies (76.9%) added triiodothyronine to the induction medium, at varying concentrations and exposure duration. Thyroid hormone is well-known to regulate the expression of many adipocyte-specific genes involved in acquisition of adipocytic phenotype, including those involved lipogenesis and lipolysis [136]. Moreover, thyroid hormone is a pivotal regulator of brown and beige adipocyte development and thermogenic function, by inducing key transcriptional programs that drive the differentiation of thermogenic adipocytes, notably through upregulation of UCP1 and PGC-1α, which are essential for mitochondrial biogenesis, oxidative metabolism, mitochondrial uncoupling, and heat generation [137,138]. Thyroid hormone also increases β-adrenergic receptor expression in adipocytes, thereby potentiating sympathetic activation of thermogenesis [139].

Surprisingly, few studies have systematically examined how the absence, presence, or varying concentrations of T3 influence the acquisition of a thermogenic phenotype in adipocyte precursors during differentiation. Herbers et al. (2022) [28] evaluated five adipogenic differentiation protocols in human adipocyte progenitors derived from the stromal vascular fraction (SVF) of abdominal subcutaneous white adipose tissue (WAT). One protocol included 2 nM T3 during the seven-day induction phase of adipogenesis, but not during the maintenance phase. Although this condition did not enhance the mRNA or protein expression of thermogenic markers relative to other protocols, it significantly increased basal respiration, maximal respiration, and proton leak [28]. In contrast, Wang et al. (2018) optimized differentiation conditions for human thermogenic adipocytes derived from fetal BAT SVF and reported that 1 nM T3 during a four-day induction period significantly upregulated thermogenic gene expression [79]. Similarly, Lee et al. (2012) [121] investigated the effects of varying T3 concentrations in the induction medium of adipose-derived stem cells from infants. They observed a dose-dependent increase in thermogenic gene expression from 10 to 250 nM T3, whereas lower concentrations (0.4 and 2 nM) had no such effect [121]. Collectively, these findings suggest that there is no universally optimal T3 concentration for thermogenic adipocyte induction. The efficacy of T3 may depend on the specific progenitor population, differentiation protocol, and the presence of additional components in the medium, which varied across these studies.

Rosiglitazone was included in the adipogenic differentiation medium of most included studies (76.1%), whereas indomethacin was used in a smaller proportion of studies (25.6%). Both chemicals are well-known for their adipogenic activity by directly activating peroxisome-proliferator activated receptor gamma2 (PPARγ2) [140]. Indomethacin, additionally, promotes adipogenesis by upregulating C/EBPβ and PPARγ2 [141]. Consistent with their action to activate PPARγ2, the inclusion of either rosiglitazone or indomethacin in the differentiation medium induces the expression of thermogenic markers in human primary adipocytes to a comparable extent [142]. However, the degree to which these effects reflect physiological thermogenic differentiation remains unclear.

Notably, only a minority of the included studies employed other agents known to induce the development of thermogenic adipocytes, such as bone morphogenetic proteins, fibroblast growth factors, irisin, and its precursor FNDC5 [109,116]. These molecules activate more complex signaling pathways and often require specific progenitor types, precisely timed intervention, and additional factors to drive thermogenic programs in adipocytes. As such, they are typically used to enhance the thermogenic phenotype rather than to initiate adipogenesis. In contrast, the routine inclusion of rosiglitazone and indomethacin in differentiation media likely reflects both their dual ability to support adipogenesis and the thermogenic phenotype in culture, in addition to their practical advantages such as lower cost and greater chemical stability compared with recombinant proteins.

Several studies also included biotin (31.6%), pantothenate (29.1%), and the iron-binding protein transferrin (43.6%) in the adipogenic differentiation medium. While biotin, pantothenate, and iron are known to influence overall adipogenesis and mitochondrial function, their specific roles in regulating thermogenic differentiation remain underexplored. Most studies have focused on their general metabolic functions, and direct mechanistic insights into their contribution to thermogenic programming are limited. Nevertheless, given the essential roles of these micronutrients in cellular metabolism, it is plausible that they also support thermogenic adipocyte development and function.

Biotin (vitamin B5) has long been recognized for its role in promoting lipid synthesis during adipogenesis by regulating acetyl-CoA carboxylase activity [143]. More recently, biotin was shown to enhance thermogenic gene expression in human primary brown adipocytes derived from the stromal vascular fraction of fetal brown adipose tissue [79]. Pantothenate (vitamin B7) similarly upregulates thermogenic markers in this model [79]. Notably, pantothenate exhibits concentration-dependent effects: at lower doses, it increases thermogenic gene expression, proton leak-associated mitochondrial respiration, and expression of genes related to the futile creatine cycle; however, higher concentrations reverse these effects and suppress glycolysis [144]. Collectively, these findings highlight the importance of biotin and pantothenate in promoting the acquisition of thermogenic characteristics during adipocyte differentiation in culture.

Transferrin is a glycoprotein with pleiotropic functions, most notably facilitating iron transport and cellular iron uptake. Iron, in turn, is essential for numerous physiological processes, particularly mitochondrial function [145]. In murine 3T3-L1 preadipocytes, iron deficiency—either through transferrin gene silencing or pharmacological chelation with deferoxamine—has been shown to impair adipogenesis [146]. Notably, brown adipogenesis appears to require greater iron availability than white adipogenesis, coinciding with increased expression of transferrin receptor 1 (TfR1) during brown, but not white, adipocyte differentiation [147]. Silencing of TfR1 disrupted the acquisition of thermogenic features in brown adipocytes, suggesting a direct link between iron uptake and thermogenic programming. Similar results were reported by Qiu et al. (2020), who further identified TfR1 as a functional marker of thermogenic adipocytes and demonstrated its essential role in maintaining mitochondrial function both in vitro and in vivo [52]. These findings suggest that transferrin may act synergistically with other components of the differentiation medium to support the induction of thermogenic phenotypes in cultured adipocytes.

Adrenergic stimulation is a well-established approach, both in vivo and in vitro, for recruiting and activating beige and brown adipocytes to induce thermogenesis. Cold exposure, the primary physiological trigger for thermogenic activation, stimulates the sympathetic nervous system to release norepinephrine, which binds to β-adrenergic receptors on brown and beige adipocytes. This signaling cascade enhances the expression of thermogenic genes and promotes lipolysis, liberating fatty acids that serve as allosteric activators of UCP1 [148]. Despite this well-characterized mechanism, only a small subset of the included studies employed pharmacological adrenergic stimulation—either short-term or prolonged—to promote thermogenic differentiation. Similarly, few studies used mild hypothermic conditions to induce thermogenic phenotypes in culture. Interestingly, previous research has shown that adipogenic precursors derived from white, but not brown, adipose tissue exhibit cold-induced thermogenic gene expression through mechanisms that appear to be independent of β-adrenergic signaling [78,149].

Adipocyte differentiation media are commonly supplemented with serum, typically fetal bovine serum (FBS), due to its content of growth factors, hormones, and nutrients that support adipogenesis [150]. Consistently, most of the studies included in this review employed serum during differentiation. However, a notable subset of studies achieved thermogenic adipocyte differentiation under serum-free conditions, as indicated by the expression of mRNA or protein thermogenesis-related markers, mitochondrial content, or mitochondrial bioenergetics. These findings suggest that chemically defined media, when supplemented with appropriate factors, can effectively support both adipogenic and thermogenic programming in primary human adipocyte precursors. The ability to induce thermogenic phenotypes in the absence of serum offers significant advantages, including improved reproducibility, reduced batch variability, and more precise control over experimental conditions. This supports the growing interest in serum-free systems for mechanistic and translational studies of human thermogenic adipocytes.

We found that thermogenic adipocytes were successfully obtained, as indicated by the expression of specific molecular markers in most studies, by the induction of precursors such as cells from the stromal vascular fraction, mesenchymal cells, and preadipocytes, despite the variations with respect to cell confluency at induction, the specific cell culture medium used, the total duration of differentiation, or the fact that differentiated was conducted in one or two or more states. Very few studies used other precursors, including adipocytes from ceiling culture, microvascular fragments, dermal fibroblasts, or progenitor cells from adipose tissue explants. This implies that precursors obtained from adipose tissue, bone marrow, or umbilical cord can be driven to express thermogenic markers when induced to differentiate with the proper ingredients.

Moreover, despite the scarcity of studies directly comparing the thermogenic potential precursors from different sources differentiated into adipocytes, the studies assessing precursors from deep neck, supraclavicular, perirenal, or periadrenal adipose tissue consistently indicated that they exhibited higher thermogenic capacity—evidenced by higher expression of thermogenesis-related markers, mitochondrial content, and mitochondrial bioenergetics—than those obtained from superficial subcutaneous or visceral depots. These findings align with imaging and histological studies identifying thermogenically active adipose regions in adult humans, particularly in the cervical and thoracic areas [131,132,133].

The most frequently reported outcomes used to assess thermogenic adipocyte differentiation in the included studies were the expression of canonical thermogenic markers at the mRNA and protein levels, along with markers distinguishing beige from brown adipocyte phenotypes. While these molecular signatures offer valuable insights into the differentiation state of adipocytes, they provide only a partial view of thermogenic identity. Notably, relatively few studies assessed mitochondrial content or function—parameters more directly linked to thermogenic capacity.

Only a minority of studies included in this review assessed mitochondrial respiration to functionally characterize thermogenic adipocytes, most commonly using extracellular flux analysis (Seahorse XF technology) to measure the oxygen consumption rate (OCR) as a proxy for mitochondrial activity. This methodology offers a valuable real-time assessment of cellular bioenergetics, allowing for the evaluation of basal respiration, proton leak, ATP-linked respiration, and maximal respiratory capacity [151,152]. In the context of thermogenic adipocytes, these parameters can provide functional validation of uncoupled respiration and UCP1 activity, which are not captured by gene or protein expression alone [153]. However, Seahorse assays also have limitations, including variability introduced by differences in cell seeding density, differentiation efficiency, and sensitivity to experimental conditions [154]. Moreover, OCR measurements must be interpreted with caution, as they do not directly quantify heat production and may reflect both UCP1-dependent and -independent pathways.

Given that thermogenic activity is fundamentally driven by mitochondrial uncoupling and oxidative metabolism, functional assays of mitochondrial bioenergentics are essential for distinguishing truly thermogenic adipocytes from those merely expressing associated markers [155]. Given the need for multi-level validation of thermogenic activity, future studies should integrate gene expression, protein abundance, and functional bioenergetic assays to more comprehensively characterize thermogenic potential and enhance the translational relevance of in vitro models.

Moreover, the physiological relevance of thermogenic adipocytes extends beyond their role in heat production. These cells are increasingly recognized for their endocrine and metabolic functions, including the secretion of lipid and protein mediators that influence systemic energy balance [128,155], as well as their capacity to act as metabolic sinks for glucose, fatty acids, and amino acids—functions that contribute to improved glucose tolerance and lipid clearance [128]. Therefore, a comprehensive in vitro characterization should go beyond classical thermogenic endpoints to include assessments of mitochondrial function, substrate utilization, and secretory activity. This broader perspective may yield more physiologically relevant models and enhance the translational potential of thermogenic adipocyte research.

It should be noted that the heterogeneity in precursor cell sources, differentiation protocols, and phenotypic characterization of thermogenic adipocyte across studies limits both direct comparisons and the synthesis of results in meta-analyses. This variability highlights the urgent need for greater standardization in protocols for differentiating and validating human thermogenic adipocytes. Despite the promising therapeutic potential of thermogenic adipocytes for obesity and its associated metabolic disorders, significant challenges remain in translating in vitro findings into clinical applications. Human cell-based models, while invaluable for mechanistic insights, lack the complex endocrine, neural, immune, and vascular signals present in vivo that are essential for thermogenic regulation [128]. Moreover, differentiation efficiency, functional maturation, and phenotypic stability of beige and brown adipocytes vary substantially depending on the cell source and protocol, which can compromise reproducibility and translational relevance [99,156].

To advance the translational relevance of human thermogenic adipocyte research, future studies should prioritize protocol standardization and comprehensive functional validation. First, clearly defining and reporting the anatomical origin, donor characteristics (e.g., age, sex, BMI), and passage number of precursor cells is essential to improve reproducibility and interpretability across studies. Second, harmonizing differentiation protocols—including media composition, duration of differentiation induction and maintenance, and exposure to key inducers such as PPARγ agonists, thyroid hormone, or cAMP induction/analogs—would enable more meaningful comparisons. Third, adopting a multi-tiered validation framework is critical: beyond mRNA and protein expression of thermogenic markers, studies should routinely incorporate functional assays such as mitochondrial respiration measurements. Moreover, adopting existing reporting standards with adaptations, such as those proposed by the Science in Risk Assessment and Policy [157] or the Organization for Economic Co-operation and Development guidance for in vitro methods [158], would further strengthen methodological transparency and data quality. Collectively, these steps would improve consistency, comparability, and translational potential across the field. Finally, given their superior ability to replicate the native adipose tissue environment, three-dimensional (3D) models should be more widely adopted as a standard approach for studying thermogenic human adipocytes. Integrating 3D and organoid systems into thermogenic research would significantly enhance the physiological relevance of in vitro studies by more accurately recapitulating critical features such as extracellular matrix composition, spatial architecture, and cell–cell interactions that influence both adipogenesis and thermogenic function.

## 5. Conclusions

This review underscores the feasibility of differentiating thermogenic adipocytes from primary human precursors using a range of two-dimensional culture protocols. While most studies employed classical adipogenic inducers alongside frequent use of triiodothyronine, rosiglitazone, or indomethacin, relatively few incorporated more complex thermogenic stimuli such as BMPs, FGFs, or irisin. Some of the included studies demonstrated thermogenic adipocyte differentiation under serum-free, chemically defined conditions, supporting their relevance for reproducible and translational applications. However, a major limitation remains the predominant reliance on gene expression as the primary outcome, with comparatively few studies evaluating mitochondrial respiration or broader aspects of metabolic function. Going forward, the development and adoption of standardized, functionally validated protocols will be critical to fully realize the potential of human in vitro thermogenic adipocyte models in metabolic research and therapeutic innovation.

## Figures and Tables

**Figure 1 cells-14-01907-f001:**
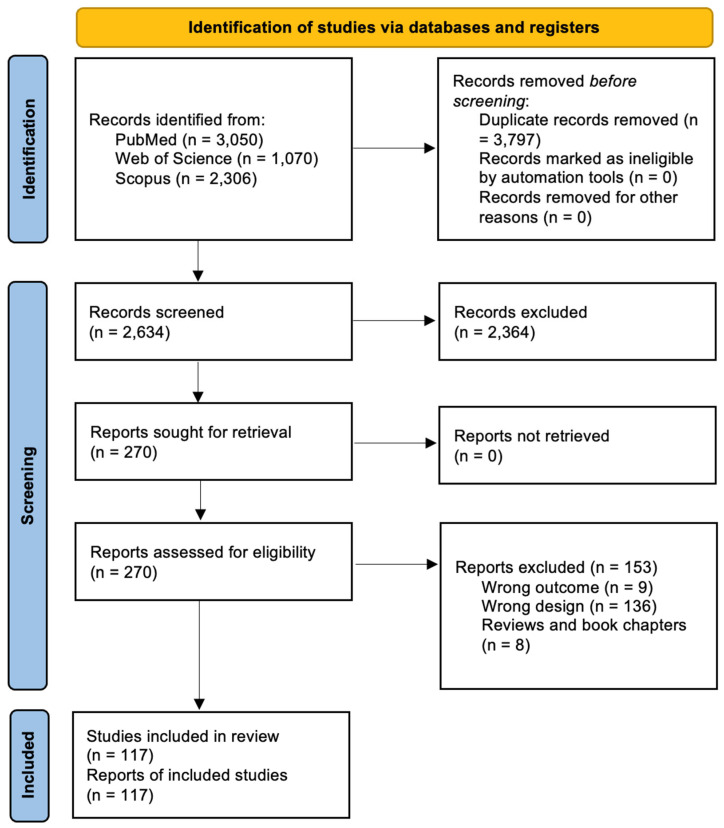
Flow diagram of the study selection process.

**Figure 2 cells-14-01907-f002:**
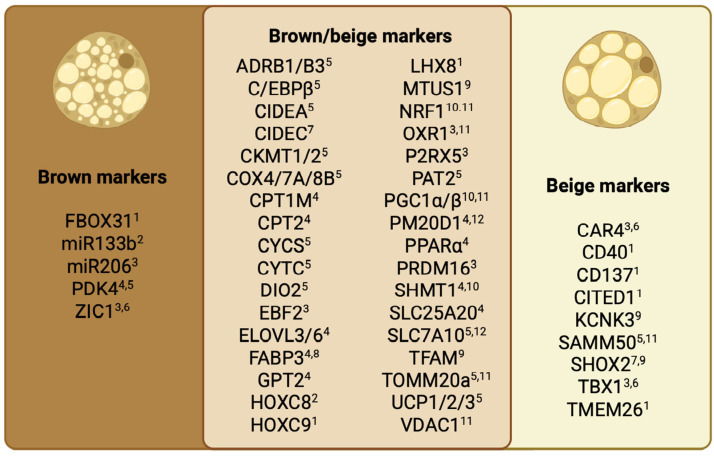
Diagram of brown, beige, and overall thermogenesis (overlapping brown/beige) RNA and protein markers assessed in the included studies. The markers are categorized as (1) having their function not defined despite being highly expressed in the indicated cells, or involved in (2) negative regulation of adipocyte type differentiation, (3) positive regulation of adipocyte type differentiation, (4) lipid metabolism, (5) thermogenesis, (6) cell function, (7) negative regulation of lipolysis, (8) lipid transport, (9) negative regulation of thermogenesis, (10) mitochondrial biogenesis, (11) mitochondrial function, (12) amino acid metabolism. Created with Biorender.com’.

**Table 1 cells-14-01907-t001:** Precursor origin and characteristics of thermogenic adipocyte differentiation conditions described in the included studies.

Characteristics ^1^	No of Studies (%)
Precursor origin	
SVF	70/117 (59.8)
Abdominal SC AT	25 (35.7)
SC WAT	21 (30)
Deep neck AT	8 (11.4)
Subcutanous neck AT	7 (10)
Supraclavicular AT	6 (8.6)
Embryonic/fetal AT	5 7.1)
Omental WAT	4 (5.7)
Breast AT	4 (5.7)
Perirenal AT	4 (5.7)
Pericardial AT	3 (4.3)
Periadrenal AT	1 (1.4)
Mediastinal AT	1 (1.4)
Mixture of AT depots	1 (1.4)
Other (chin, thigh, knee, and subacromial AT)	4 (5.7)
WAT (depot NR)	1 (1.4)
NR	3 (4.3)
ADSC	16 (13.7)
MADSC	11 (9.4)
Stromal cells	8 (6.8)
BM	6 (75)
Umbilical cord	1 (12.5)
WAT SC	1 (12.5)
Preadipocytes	6 (5.1)
MSCs	5 (4.3)
Other ^2^	9 (7.7)
Confluency at induction	
<100%	14 (12.8)
At confluency	48 (41)
After confluency	
2 days	26 (22.2)
1 day	1 (0.9)
NR	28 (23.9)
FGF for proliferative phase (before induction)	35 (29.9)
Medium for induction	
DMEM	41 (35)
Serum free	1 (2.4)
FBS	32 (78)
Other type of serum (NCS, FCS)	8 (19.5)
DMEM-F12	42 (35.9)
Serum free	21 (50)
FBS	16 (38.1)
Other type of serum (human, FCS)	5 (11.9)
DMEM-F12-Ham’s	20 (17.1)
Serum free	15 (75)
FBS	4 (20)
Other (FCS)	1 (5)
MEMα	4 (3.4)
Serum free	0 (0)
FBS	4 (100)
Commercial medium	2 (1.2)
NR	5 (4.3)
Other ^3^	4 (3.4)
Differentiation period	
<7 days	5 (4.3)
7–14 days	67 (57.3)
>14 days	38 (32.5)
Other ranges	6 (5.1)
NR	2 (1.7)
Differentiation stages	
One	35 (29.9)
Two	65 (55.6)
Three or more	17 (14.5)

^1^ As some studies reported more than one outcome/measure, the total percentage exceeds 100%. ^2^ Adipocytes from ceiling culture, microvascular fragments, dermal fibroblasts, progenitor from adipose tissue explants, sprouting cells from cultured AT, endothelial/capillary network cells. ^3^ IMDM, DMEM/DMEM-F12, DMEM-F10, DMEM-F10-Ham’s. ADSCs: Adipose-derived stem cells; AT: adipose tissue; DMEM: Dulbecco’s Modified Eagle’s Medium; DMEM-F12: Dulbecco’s Modified Eagle’s Medium/Nutrient F-12; FBS: fetal bovine serum; FCS: fetal calf serum; FGF: fibroblast growth factor; IMDM: Iscove’s modified Dulbecco’s medium; MADSCs: multipotent adipose-derived stem cells; MEM: Minimum Essential Medium; MSC: mesenchymal stem cells; NCS: neonatal calf serum; NR: not reported; SC: subcutaneous; WAT: white adipose tissue.

**Table 2 cells-14-01907-t002:** Characteristics of induction medium to promote thermogenic adipocyte differentiation in the included studies.

Characteristics	No of Studies (%)
IBMX—no (%)	117 (100)
No IBMX—no (%)	9 (7.7)
IMBX at induction—no (%)	82 (70.1)
Concentration	0.1–1 mM
Duration	2–14 days
IMBX during whole differentiation—no (%)	26 (22.2)
Concentration	0.03–0.5 mM
Duration	4–32 days
Dexamethasone—no (%)	117 (100)
At induction	71 (60.7)
Concentration	0.1–25 μM
Duration	2–14 days
During whole differentiation	46 (39.3)
Concentration	0.1–100 μM
Duration	7–32 days
Insulin—no (%)	114 (97.4)
At induction—no (%)	16 (14.0)
Concentration	1–1000 nM
Duration	1–14 days
During whole differentiation—no (%)	98 (86.0)
Concentration	1–1000 nM
Duration	7–32 days
T3—no (%)	90 (76.9)
At induction—no (%)	13 (14.4)
Concentration	0.2–250 nM
Duration	3–10 days
During whole differentiation—no (%)	73 (81.1)
Concentration	0.2–1000 nM
Duration	6–32 days
After induction—no (%)	4 (4.4)
Concentration	0.2–1 nM
Duration	6–10 days
Other browning agents—no (%)	18 (15.3)
BMP4 or 7—no (%)	12 (66.7)
FGF—no (%)	4 (22.2)
Irisin—no (%)	1 (5.5)
FNDC5—no (%)	1 (5.5)
Rosiglitazone—no (%)	89 (76.1)
At induction—no (%)	25 (28.1)
Concentration	0.2–5 μM
Duration	3–14 days
During whole differentiation—no (%)	41 (46.1)
Concentration	0.1–10 μM
Duration	3–32 days
At different time points—no (%)	23 (25.8)
Concentration	0.1–1 μM
Indomethacin—no (%)	30 (25.6)
Concentration	2–200 μM
At induction—no (%)	18 (60)
During whole differentiation—no (%)	12 (40)
Transferin—no (%)	51 (43.6)
Concentration	6.25–10 μg/mL
At induction—no (%)	10 (19.6)
During whole differentiation—no (%)	40 (78.4)
After induction—no (%)	1 (2)
Biotin—no (%)	37 (31.6)
Concentration	3.3–330 μM
At induction—no (%)	4 (10.8)
During whole differentiation—no (%)	33 (89.2)
Pantothenate—no (%)	34 (29.1)
Concentration	17–20 μM
At induction—no (%)	4 (11.8)
During whole differentiation—no (%)	30 (88.2)
Adrenergic stimulation	
NE—no (%)	12 (10.3)
Concentration	1–10 μM
Acute (<12 h)—no (%)	10 (83.3)
Chronic (>12 h)—no (%)	2 (16.7)
Forskolin—no (%)	12 (10.3)
Concentration	1–50 μM
Acute (<12 h)—no (%)	6 (50)
Chronic (>12 h)—no (%)	6 (50)
CL 316,243—no (%)	9 (7.7)
Concentration	1–10 μM
Acute (<12 h)—no (%)	3 (33.3)
Chronic (>12 h)—no (%)	6 (66.7)
Isoproterenol	6 (5.1)
Concentration	0.1–100 μM
Acute (<12 h)—no (%)	4 (66.7)
Chronic (>12 h)—no (%)	1 16.7)
NR	1 (16.7)
cAMP—no (%)	4 (3.4)
Concentration	0.5–1 mM
Acute (<12 h)—no (%)	4 (100)
Chronic (>12 h)—no (%)	1 (25)
Cold exposure—no (%)	6 (5.1)
Temperature range	16–32 °C
Acute (<12 h)	5 (83.3)
Chronic (>12 h)	1 (16.7)

As some studies reported more than one outcome/measure, the total percentage exceeds 100%. BMP: bone morphogenetic protein; cAMP: cyclic adenosine monophosphate; IBMX: isomethylbuthylxanthine; FGF: fibroblast growth factor; FNDC5: fibronectin type III domain-containing protein 5; NR: not reported; T3: triiodothyronine.

**Table 3 cells-14-01907-t003:** Characterization of thermogenic adipocyte in the included studies.

Characteristics	No of Studies (%)
Outcomes	
RNA expression (thermogenic markers)—no (%)	104/117 (88.9)
qPCR—no (%)	102/104 (98.1)
Transcriptome—no (%)	3/104 (2.9)
Protein expression (thermogenic markers)—no (%)	70/117 (59.8)
Mitochondrial bioenergetics—no (%)	48/117 (41)
Mitochondrial content—no (%)	21/117 (17.9)
Comparison	108/117 (92.3)
Thermogenic vs. white adipocytes—no (%)	43/108 (39.8)
↑ thermogenic markers (RNA)	35/38 (92.1)
↑ thermogenic markers (protein)	25/28 (89.3)
↑ mitochondrial activity	10/12 (83.3)
↑ mitochondrial content	7/8 (87.5)
Stimulated vs. non-stimulated with adrenergic agonists/FSK/cAMP—no (%)	28/108 (25.9)
↑ thermogenic markers (RNA)	21/28 (75)
↑ thermogenic markers (protein)	8/9 (88.9)
↑ mitochondrial activity	14/17 (82.4)
↑ mitochondrial content	2/2 (100)
Thermogenic adipocytes vs. undifferentiated cells—no (%)	25/108
↑ thermogenic markers (RNA)	19/21 (90.5)
↑ thermogenic markers (protein)	12/14 (85.7)
↑ mitochondrial activity	1/1 (100)
↑ mitochondrial content	4/4 (100)
Thermogenic adipocytes differentiated from precursors from different AT depots—no (%)	19/117 (16.2)
Deep cervical vs. superficial cervical and other subcutaneous AT depots—no (%)	6/19 (31.6)
↑ thermogenic markers (RNA)	4/4 (100)
↑ thermogenic markers (protein)	1/1 (100)
↑ mitochondrial activity	2/2 (100
↑ mitochondrial content	-
Supraclavicular vs. other subcutaneous AT depots—no (%)	4/19 (21.1)
↑ thermogenic markers (RNA)	3/3 (100)
↑ thermogenic markers (protein)	2/2 (100
↑ mitochondrial activity	-
↑ mitochondrial content	-
Periadrenal/perirenal vs. other subcutaneous AT depots—no (%)	3/19 (15.8)
↑ thermogenic markers (RNA)	2/3 (66.7)
↑ thermogenic markers (protein)	1/1 (100)
↑ mitochondrial activity	1/1 (100)
↑ mitochondrial content	-
Fetal/embryonic vs. adult AT depots—no (%)	3/19 (15.8)
↑ thermogenic markers (RNA)	3/3 (100)
↑ thermogenic markers (protein)	2/2 (100)
↑ mitochondrial activity	1/1 (100)
↑ mitochondrial content	2/2 (100)
Cells not treated with glitazones—no (%)	9/107 (8.4)
↑ thermogenic markers (RNA)	6/7 (85.7)
↑ thermogenic markers (protein)	4/4 (100)
↑ mitochondrial activity	2/2 (100)
↑ mitochondrial content	2/2 (100)
Adipocytes exposed to lower vs. higher temperature—no (%)	6/107 (5.6)
↑ thermogenic markers (RNA)	2/4 (50)
↑ thermogenic markers (protein)	3/3 (100)
↑ mitochondrial activity	-
↑ mitochondrial content	-

As some studies reported more than one outcome/measure, the total percentage exceeds 100%. RNA: ribonucleic acid; qPCR: quantitative real-time polymerase chain reaction; AT: adipose tissue.

## Data Availability

All data supporting the findings of this study are available within the manuscript and its Appendix A.

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
