# Peer review of "Thermogenic Differentiation of Human Adipocyte Precursors in Culture: A Systematic Review"

_cells, 2025, doi:10.3390/cells14231907_

Round 1

Reviewer 1 Report

Comments and Suggestions for Authors

This systematic review presents a comprehensive synthesis of the current literature on human in vitro models for thermogenic adipocyte induction. The manuscript is well-structured, and the results are valuable for the field of metabolic research, particularly regarding the translational potential of thermogenic adipocytes in combating metabolic diseases. The review demonstrates that the authors conducted a thorough literature search and critically appraised the included studies. However, some aspects require clarification and further elaboration for improved rigor and impact. Specifically, the manuscript would benefit from more detailed discussion on the limitations of current in vitro models, standardization of outcome measures, and the translational relevance of the findings. Addressing these points would enhance the overall quality and utility of the review.

Introduction

  1. The introduction highlights the limitations of murine models for studying thermogenic adipocytes. Could the authors elaborate on the specific physiological differences between murine and human adipose tissue that most significantly impact translational research?

  2. The authors mention the therapeutic potential of thermogenic adipocytes for metabolic diseases. What are the main challenges in translating findings from in vitro human models to clinical applications in humans?

Materials and Methods

  1. The review states that studies using mouse cells or cell lines were excluded. What was the rationale for excluding studies that used immortalized human adipocyte cell lines, and how might their inclusion have affected the conclusions?

  2. The authors used the NTP/OHAT risk of bias tool for quality appraisal. Were any additional criteria or tools considered to assess the methodological quality of in vitro studies, given their unique challenges compared to in vivo research?

Results

  1. The majority of studies relied on gene expression as the primary outcome. How did the authors address the potential limitations of using gene expression alone to define the thermogenic phenotype?

  2. The review notes that only a minority of studies assessed mitochondrial respiration or metabolic function. Can the authors provide more detail on the methodologies used in those studies and their relative strengths and weaknesses?

  3. The included studies used a wide range of precursor cell types and differentiation protocols. How did the authors account for this heterogeneity when synthesizing the results?

  4. Several studies demonstrated successful differentiation under serum-free conditions. What criteria were used to define "successful" thermogenic differentiation in these cases?

  5. The review mentions depot-specific differences in thermogenic potential. Were there any consistent patterns or key findings regarding which adipose depots yielded the most robust thermogenic adipocytes?

Discussion

  1. The discussion emphasizes the need for standardized protocols and functional validation. What specific recommendations can the authors provide for future studies to achieve greater standardization in this field?

  2. The review briefly discusses the role of biotin, pantothenate, and transferrin in thermogenic differentiation. Could the authors expand on the potential mechanisms by which these factors influence thermogenic programming?

  3. Given the predominance of 2D culture systems in the included studies, how might the adoption of 3D or organoid models impact the study of human thermogenic adipocytes and their translational relevance?

Author Response

Comment 1: The introduction highlights the limitations of murine models for studying thermogenic adipocytes. Could the authors elaborate on the specific physiological differences between murine and human adipose tissue that most significantly impact translational research?

Response 1: Thank you very much for this thoughtful comment. We have briefly addressed the limitations of murine models for studying thermogenic adipocytes in both the abstract and the discussion. In the introduction, we focused primarily on the current challenges in obesity treatment to justify the rationale for investigating energy expenditure as a potential therapeutic avenue. To maintain clarity and conciseness and address the reviewer’s concern, we chose to elaborate on species-specific differences between murine and human adipose tissue—particularly those that impact translational relevance—in the discussion section. However, if the reviewers feel that including this content in the introduction would improve the manuscript, we will gladly make that adjustment. Please see lines 251-268 of the revised manuscript.

Comment 2: The authors mention the therapeutic potential of thermogenic adipocytes for metabolic diseases. What are the main challenges in translating findings from in vitro human models to clinical applications in humans?

Response 2: We thank the reviewer for this thoughtful question. We apologize for not including this discussion in the introduction. To maintain focus and avoid extending the introduction, we chose to address the translational challenges associated with in vitro human models in the discussion section of the manuscript, as a limitation of the review. There, we highlight that although in vitro systems - such as primary human adipocyte cultures or stem-cell-derived thermogenic adipocytes - have provided valuable mechanistic insights, several limitations constrain their translational potential. These include the absence of key regulatory inputs present in vivo (e.g., endocrine, neural, and immune signals), variability in differentiation efficiency and functional stability across cell models. If the reviewer prefers, we would be happy to integrate a summary of these translational limitations into the introduction to better frame the scope and relevance of our study. Please see lines 459-470 of the revised manuscript.

Comment 3: The review states that studies using mouse cells or cell lines were excluded. What was the rationale for excluding studies that used immortalized human adipocyte cell lines, and how might their inclusion have affected the conclusions?

Response 3: We appreciate the reviewer’s question regarding our inclusion criteria. We chose to exclude studies utilizing immortalized human adipocyte cell lines from our review in order to focus on models with the highest physiological relevance to human thermogenic adipocyte biology. Although immortalized cell lines are useful tools for mechanistic studies due to their ease of use, reproducibility, and scalability, their thermogenic and even adipogenic potential may differ significantly from that of primary cells. Immortalized lines may exhibit different transcriptional responses, receptor expression patterns, and metabolic activity due to the process of immortalization or prolonged passaging. These aspects can influence the interpretation of results, particularly in studies aimed at modeling in vivo human thermogenic function or evaluating translational potential.

We completely agree that including data from immortalized human adipocyte lines could have broadened the scope of findings but could also have introduced variability not representative of native human adipose tissue. By limiting our review to primary cells and more physiologically relevant models, we aimed to provide a clearer picture of mechanisms with direct relevance to human biology and therapeutic translation. That said, we recognize that immortalized lines remain valuable in specific experimental contexts, and we have added a brief comment to the discussion acknowledging their role and potential limitations, should the reviewer find that helpful. Please see lines 98-103 of the revised manuscript.

Comment 4: The authors used the NTP/OHAT risk of bias tool for quality appraisal. Were any additional criteria or tools considered to assess the methodological quality of in vitro studies, given their unique challenges compared to in vivo research?

Response 4: We thank the reviewer for this important observation. We selected the NTP/OHAT risk of bias tool as our primary framework for quality appraisal because it offers a structured and transparent approach for assessing internal validity across a broad range of study types, including in vitro experiments. The tool has been adapted to capture sources of bias relevant to non-animal studies, such as selection, performance, and detection bias, which are also pertinent in in vitro contexts.

We acknowledge that additional in vitro–specific tools exist, such as SciRAP (Science in Risk Assessment and Policy) and ToxRTool (Toxicological data Reliability Assessment Tool), which were developed to evaluate the methodological quality and reporting of in vitro toxicological studies. While these frameworks were not systematically applied in our review in order to maintain consistency with prior OHAT-based assessments, we incorporated several of their principles—particularly regarding the reporting of cell identity, differentiation protocols, and validation of functional endpoints—into our qualitative appraisal.

Should the reviewers recommend the application of multiple tools for quality assessment, we would be happy to incorporate them in a revised analysis.

Comment 5: The majority of studies relied on gene expression as the primary outcome. How did the authors address the potential limitations of using gene expression alone to define the thermogenic phenotype?

Response 5: We thank the reviewer for this important point. We acknowledge that the majority of studies included in our review relied primarily on gene expression analyses, particularly the measurement of the mRNA levels of overall thermogenic markers, such as UCP1 and PPARGC1A, to define the thermogenic phenotype. While UCP1 is a hallmark gene for thermogenic adipocytes, gene expression alone does not fully capture functional thermogenic capacity, which is ultimately dependent on mitochondrial activity and uncoupled respiration. To address this limitation, we critically evaluated whether each study included complementary assessments - such as protein expression, mitochondrial content, and mitochondrial function assays, such as oxygen consumption rate - and noted these distinctions in our qualitative synthesis. We also highlighted in the discussion the need for functional validation of thermogenesis in future studies, as reliance on mRNA expression alone may lead to overestimation or misinterpretation of thermogenic capacity. We emphasized the importance of integrating multi-level readouts to more accurately define the thermogenic phenotype in vitro. We agree with the reviewer that strengthening the field's reliance on functional endpoints will be essential to improving the translational value of in vitro thermogenesis models and tried to address this in the discussion and in the conclusion. Please see lines 422-448 of the revised manuscript.

Comment 6: The review notes that only a minority of studies assessed mitochondrial respiration or metabolic function. Can the authors provide more detail on the methodologies used in those studies and their relative strengths and weaknesses?

Response 6: We are thankful for the comment and have expanded information on the assessment of mitochondrial bionenergetics in the included studies, in the Discussion Section, as follows (please see lines 429-441 in the revised):

Only a minority of studies included in this review assessed mitochondrial respiration to functionally characterize thermogenic adipocytes, most commonly using extracellular flux analysis (Seahorse XF technology) to measure the oxygen consumption rate (OCR) as a proxy for mitochondrial activity. This methodology offers a valuable real-time assessment of cellular bioenergetics, allowing for the evaluation of basal respiration, proton leak, ATP-linked respiration, and maximal respiratory capacity [152,153]. In the context of thermogenic adipocytes, these parameters can provide functional validation of uncoupled respiration and UCP1 activity, which are not captured by gene or protein expression alone [154]. However, Seahorse assays also have limitations, including variability introduced by differences in cell seeding density, differentiation efficiency, and sensitivity to experimental conditions [155]. Moreover, OCR measurements must be interpreted with caution, as they do not directly quantify heat production and may reflect both UCP1-dependent and -independent pathways.

Comment 7: The included studies used a wide range of precursor cell types and differentiation protocols. How did the authors account for this heterogeneity when synthesizing the results?

Response 7: The included studies exhibited substantial heterogeneity in both the source of precursor cells—ranging from subcutaneous and deep neck fat depots to mesenchymal stromal cells—and in the differentiation protocols used to induce thermogenic adipocytes. This variability represents a critical challenge for data synthesis, as differences in precursor identity, donor characteristics, and culture conditions can all influence adipogenic and thermogenic potential. To address this, we performed a qualitative synthesis rather than a meta-analysis, explicitly noting the precursor type, depot origin, and key features of the differentiation protocols for each study. We expanded our discussion to address the reviewer’s concern, emphasing how these variables may contribute to discrepancies in thermogenic marker expression and functional outcomes. While this heterogeneity limits direct comparison across studies, it also reflects the diverse approaches currently used in the field and underscores the need for greater standardization in protocols for differentiating and validating human thermogenic adipocytes. Please see lines 459-470 in the revised manuscript.

Comment 8: Several studies demonstrated successful differentiation under serum-free conditions. What criteria were used to define "successful" thermogenic differentiation in these cases?

Response 8: In studies employing serum-free differentiation protocols, “successful” thermogenic differentiation was typically defined using a combination of morphological, molecular, and functional criteria. Most commonly, success was indicated by the upregulation of thermogenic genes mRNA levels. A subset of studies further supported thermogenic identity through UCP1 or other thermogenesis-related protein detection, or functional assessments of mitochondrial respiration, such as oxygen consumption rate measurements. For the purposes of this review, we considered the minimum criterion for thermogenic differentiation to be the induction of thermogenesis-related mRNA expression, acknowledging that this represents a limited but commonly reported endpoint. When available, we documented the presence of additional validation measures, including protein-level analysis and metabolic function, and highlighted these in our synthesis to provide context on the robustness of thermogenic characterization across studies. We have detailed this in the Methods section and also in the Results and Discussion sections. Please see lines 80-82 in the Methods section, lines 492-497 in the Discussion section, and lines 499-504 of the Conclusions section of the revised manuscript. We have also removed the word successful because we view that it is a strong word.

Comment 9: The review mentions depot-specific differences in thermogenic potential. Were there any consistent patterns or key findings regarding which adipose depots yielded the most robust thermogenic adipocytes?

Response 9: We are thankful for the comment and to address the reviewer’s concern we have expanded the results presented in Table 3 in the Results and Discussion sections. The review revealed depot-specific differences in thermogenic potential among human adipocyte precursors. Across all twelve studies comparing adult fat depots, precursors derived from deep neck, supraclavicular, perirenal, or periadrenal adipose tissue consistently exhibited higher potential to differentiate into thermogenic adipocytes - evidenced by greater thermogenesis-related markers, mitochondrial content, and mitochondrial bioenergetics - than those obtained from superficial subcutaneous or visceral depots. These findings align with imaging and histological studies identifying thermogenically active adipose regions in adult humans, particularly in the cervical and thoracic areas. Although methodological heterogeneity limited quantitative synthesis, the convergence of results across multiple studies supports the notion that adipose tissue from specific anatomical sites harbors an intrinsically higher propensity for thermogenic differentiation. Please see lines 220-227 in the Results section and lines 413-421 of the Discussion section in the revised manuscript.

Comment 10: The discussion emphasizes the need for standardized protocols and functional validation. What specific recommendations can the authors provide for future studies to achieve greater standardization in this field?

Response 10: We thank the reviewer for this thoughtful comment. In response, we have added a paragraph to the Discussion section outlining specific recommendations to advance standardization in the field, based on the findings of our review. Please see lines 471-486 of the revised manuscript.

Comment 11: The review briefly discusses the role of biotin, pantothenate, and transferrin in thermogenic differentiation. Could the authors expand on the potential mechanisms by which these factors influence thermogenic programming?

Response 11: We appreciate the reviewer’s comment in the roles of biotin, pantothenate, and transferrin in thermogenic adipocyte differentiation. These factors were components of the differentiation media in the included studies and are well known to contribute to cellular metabolism and mitochondrial function. However, their specific roles in regulating thermogenic differentiation remain underexplored. Most studies have focused on their general metabolic functions, and direct mechanistic insights into their contribution to thermogenic programming are limited. To address the reviewer’s concern, we included this information in the Discussion section. Please see lines 344-351 of the revised manuscript.

Comment 12: Given the predominance of 2D culture systems in the included studies, how might the adoption of 3D or organoid models impact the study of human thermogenic adipocytes and their translational relevance?

Response 12: We are grateful for the insightful comment raised by the reviewer. Although our review did not restrict the search to studies using 2D models, we pre-specified the exclusion of 3D culture systems to maintain methodological consistency across included studies. This decision was guided by the current scarcity and heterogeneity of 3D models applied to human thermogenic adipocytes. Indeed, we identified only three studies utilizing organoid systems - two of which are listed in Appendix 4 (excluded studies), and the study by Yang et al. (2017), which included both 2D and 3D culture data, and was described in our review along with the other 116 included studies involving 2D systems.

While 2D models have provided critical insights into the molecular regulation of thermogenic adipocytes, they do not fully capture the structural, cellular, and microenvironmental complexity of adipose tissue in vivo. Three-dimensional and organoid systems offer distinct advantages in this regard, including improved extracellular matrix architecture, spatial organization, and cell-cell interactions that may influence both adipogenesis and plausibly thermogenic programming. Notably, the 3D component of the Yang et al. (2017) study demonstrated enhanced expression of thermogenesis-associated genes and increased proton leak-linked respiration when compared with the 2D component, underscoring the potential of such systems to more faithfully model thermogenic function. As such, 3D models may serve as valuable platforms to improve the physiological relevance and translational potential of in vitro studies, particularly for therapeutic screening and drug discovery. We have addressed this in the Discussion secion (please see lines 486-492 of the revised manuscript).

Reviewer 2 Report

Comments and Suggestions for Authors

The review manuscript entitled “ Thermogenic differentiation of adipocyte precursor in culture: a systematic review” summarizes recent studies using human cell models for thermogenic differentiation.

The summary is informative and timely. However, there are some concerns noted in the current version.

  1. It would be better to add “human” somewhere to the title to indicate that the review is about human cell model of thermogenic adipocyte differentiation.
  2. Fig 2 shows brown, brown/beige, and beige marker genes. Why are UCP1 or PGC1a not included? Additionally, it may be more informative to categorize these genes and include relevant citations.
  3. It is not clear what the main points are from lines 213-217. Also, it may be more helpful to summarize which human white fat depot shows the greatest potential for thermogenic differentiation.
  4. Appendix 3, which summarizes each included study, is helpful. However, it would be equally or more helpful to include corresponding citations to Table 1-3.
  5. There are some editorial errors found. For example, line 66 and line 172. Please carefully edit the manuscript for clarity.

Author Response

Comment 1: It would be better to add “human” somewhere to the title to indicate that the review is about human cell model of thermogenic adipocyte differentiation.

Response 1: We completetly agree and have included the word “human” in the title.

Comment 2: Fig 2 shows brown, brown/beige, and beige marker genes. Why are UCP1 or PGC1a not included? Additionally, it may be more informative to categorize these genes and include relevant citations.

Response 2: Thank you for the insightful comment. We did include UCP1 and PGC1α among the marker genes in Figure 2; however, we appologize that the figure was not clear and improved it for clarity, also mentioning in the legend that the markers are listed in alphabetical order. Regarding categorization and the inclusion of citations within the figure, we aimed to maintain clarity and visual simplicity. Adding citations directly in the figure risks cluttering the presentation and reducing readability, so we added some references to the figure legend. Additionally, categorizing the genes beyond the current grouping into brown, beige, and brown/beige markers proved challenging due to overlap in gene function and expression profiles across adipocyte types. To address the reviwer’s concern and maintain the simplicity of the figure, we have detailed the information in Appendix 4, by adding the function of the marker and the references.

Comment 3: It is not clear what the main points are from lines 213-217. Also, it may be more helpful to summarize which human white fat depot shows the greatest potential for thermogenic differentiation.

Response 3: we apologize that the main points from lines 213 to 217 are unclear. We have rewriten this part of the manuscript to improve its clarity. The review revealed depot-specific differences in thermogenic potential among human adipocyte precursors. Across all twelve studies comparing adult fat depots, precursors derived from deep neck, supraclavicular, perirenal, or periadrenal adipose tissue consistently exhibited higher potential to differentiate into thermogenic adipocytes - evidenced by greater thermogenesis-related markers, mitochondrial content, and mitochondrial bioenergetics - than those obtained from superficial subcutaneous or visceral depots. These findings align with imaging and histological studies identifying thermogenically active adipose regions in adult humans, particularly in the cervical and thoracic areas. Although methodological heterogeneity limited quantitative synthesis, the convergence of results across multiple studies supports the notion that adipose tissue from specific anatomical sites harbors an intrinsically higher propensity for thermogenic differentiation. Please see lines 220-227 in the Results section and lines 413-421 of the Discussion section in the revised manuscript.

Comment 4: Appendix 3, which summarizes each included study, is helpful. However, it would be equally or more helpful to include corresponding citations to Table 1-3.

Response 4: We appreciate the reviewer’s suggestion to include citations directly in the table. Unfortunately, we apologize that the addition of the citations to the table compromised its visual clarity and readability, since some of the data had more than 100 citations. To maintain a clean and accessible presentation of the data, we have kept the table citation-free in the main text and added the citations to Appendix 3. If the reviewer considers it helpful, we would be happy to provide a duplicate version of the table with the relevant citations included in an appendix.

Comment 5: There are some editorial errors found. For example, line 66 and line 172. Please carefully edit the manuscript for clarity.

Response 5: we apologize for these errors and have corrected them. We have also thoroughly reviewed the entire manuscript.

Round 2

Reviewer 1 Report

Comments and Suggestions for Authors

The revised version of manuscript is sufficient for publication.